# Spatial gene expression maps of the intestinal lymphoid follicle and associated epithelium identify zonated expression programs

**Noam Cohen** [1,2], **Hassan Massalha** [2,3], **Shani Ben-Moshe** [2], **Adi Egozi** [2],
**Milena Rozenberg** [2], **Keren Bahar Halpern** [2], **Shalev Itzkovitz** [2] *

**1** Department of Biotechnology, Israel Institute for Biological Research, Ness-Ziona, Israel, **2** Department of Molecular Cell Biology, Weizmann Institute of Science, Rehovot, Israel, **3** Wellcome Sanger Institute, Wellcome Genome Campus, Hinxton, Cambridge, United Kingdom

* shalev.itzkovitz@weizmann.ac.il

**Data Availability Statement:** Raw and processed sequencing data are available in the Supplementary

## Abstract

The intestine is lined with isolated lymphoid follicles (ILFs) that facilitate sampling of luminal antigens to elicit immune responses. Technical challenges related to the scarcity and small sizes of ILFs and their follicle-associated epithelium (FAE) impeded the characterization of their spatial gene expression programs. Here, we combined RNA sequencing of laser capture microdissected tissues with single-molecule transcript imaging to obtain a spatial gene expression map of the ILF and its associated FAE in the mouse small intestine. We identified zonated expression programs in both follicles and FAEs, with a decrease in enterocyte antimicrobial and absorption programs and a partial induction of expression programs normally observed at the villus tip. We further identified Lepr+ subepithelial telocytes at the FAE top, which are distinct from villus tip Lgr5+ telocytes. Our analysis exposes the epithelial and mesenchymal cell states associated with ILFs.

## Introduction

The intestine is lined with mucosa-associated lymphoid follicles that promote homeostatic response against luminal microbiota [1]. These lymphoid follicles include 7 to 10 large Peyer's patches (1 to 2 mm in diameter) [2], as well as smaller solitary intestinal lymphoid tissues (SILTs) that are up to 200 μm in diameter. SILTs consist of a spectrum of developed tissues, ranging from cryptopatches (CPs) to mature isolated lymphoid follicles (ILFs), numbering at around 100 to 200 along mouse small intestine [3,4]. Peyer's patches and mature ILFs are covered by follicle-associated epithelium (FAE), forming a dome-like shape [5]. The FAE is mainly composed of enterocytes, with a minority of interspersed Microfold (M) cells. Secretory goblet cells have been shown to be scarce along the Peyer's patch FAE, resulting in a thinner mucus layer at the luminal surface, and by that facilitating close interactions with microbial antigens [6]. The thin mucous layer enables apical–basolateral transport of antigens from the lumen across the FAE into the lamina propria by M cells, mediating the mucosal immune response [7]. This transport program is different from the normal function of the

tables and in the GenBank GEO database NCBI (GSE168483).

**Funding:** This work was funded by the Israel Science Foundation grant No. 1486/16 (S.I.); the Broad Institute-Israel Science Foundation grant No. 2615/18 (S.I.); the Wolfson Family Charitable Trust (S.I.); the Edmond de Rothschild Foundations (S.I.); the Fannie Sherr Fund (S.I.); the Dr. Beth Rom-Rymer Stem Cell Research Fund (S.I.); the Minerva Stiftung grant (S.I.); the European Research Council (ERC) under the European Union's Horizon 2020 research and innovation program grant No. 768956. (S.I.); the Bert L. and N. Kuggie Vallee Foundation (S.I.); the Howard Hughes Medical Institute (HHMI) international research scholar award (S.I.) and the Chan Zuckerberg Initiative grant No. CZF2019-002434 (S.I.) The funders had no role in study design, data collection and analysis, decision to publish, or preparation of the manuscript.

**Competing interests:** The authors have declared that no competing interests exist.

**Abbreviations:** CP, cryptopatch; FAE, follicle-associated epithelium; FAEB, FAE bottom; FAET, FAE top; ILF, isolated lymphoid follicle; ILFB, ILF bottom; ILFT, ILF top; LCM RNA-seq, laser capture microdissection RNA sequencing; LTi, lymphoid tissue–induced; SILT, solitary intestinal lymphoid tissue; smFISH, single-molecule fluorescence in situ hybridization; UMI, unique molecular identifier; VB, villus bottom.

intestinal epithelium, which consists of selective transport of nutrients with a simultaneous blocking of microbial access to the surface.

Several studies described the transcriptomes of FAE cells and M cells in Peyer's patches [8–10]; however, technical limitations related to the smaller sizes of ILFs impeded their similar transcriptomic characterization. Here, we applied laser capture microdissection RNA sequencing (LCM RNA-seq; [11,12]) and single-molecule fluorescence in situ hybridization (smFISH; [13,14]) to generate a spatial expression map of ILFs and their associated FAE in the mouse small intestine. We identify zonated expression programs in ILFs and FAE, with a decrease in antimicrobial and absorption programs at the FAE top and a compensatory increase in secreted antimicrobial peptides at the FAE boundaries. The FAE top consists of enterocytes that partially induce genes normally observed at the villus tip and are in contact with Lepr+ subepithelial telocyte cells.

## Results

### Spatial mapping of ILF and FAE gene expression

To investigate the spatial heterogeneity of epithelial, immune, and stromal gene expression in the ILF and its associated FAE, we used LCM to isolate 5 segments. These included 3 epithelial segments consisting of FAE top (FAET), FAE bottom (FAEB), and an adjacent segment at the bottom of a neighboring villus (denoted villus bottom (VB)), and 2 immune segments consisting of the ILF top (ILFT) and ILF bottom (ILFB) (Fig 1A–1C). We used mcSCRBseq [15,16], a sensitive protocol based on unique molecular identifiers (UMIs; Methods) to sequence the RNA of the dissected fragments. We identified distinct zonated expression and enriched gene sets in each of the segment (Figs 1D and S1). We used these LCMseq expression programs as a basis for extensive smFISH validations to establish their statistical power.

To assess the cellular composition of each of segment, we performed computational deconvolution using the CIBERSORTx tool [17], which included epithelial and immune single cell–based signatures ([8,11]; Fig 1E and S1 Data). Among the immune cells, Plasmablasts, B cells, neutrophils, and NK cells were more abundant at the ILFB, whereas CD8 T cells and dendritic cells were more highly represented (yet not statistically significantly) at the ILFT or the FAET. Consistently, we identified higher expression of different B cell markers, such as *Cd19*, at the bottom of the ILF (S2A, S2D and S2I Fig). In contrast, T cell markers did not show a spatial bias toward a specific zone in the ILF (S2J Fig). Rather, using smFISH, we found that *Cd3e*, *Cd4*, and *Cd8*, classic T cell markers, as well as *Gzma*, a marker of cytotoxic CD8+ T cells, were highly abundant at the ILFT. (S2B, S2C and S2E–S2H Fig). Consistently, CD8+ and Gzma+ cells were abundantly intercalated across the FAET (S3A Fig), whereas CD4+ cells (S2G Fig) as well as the Treg marker Ctla4 were zonated toward the ILFT (S3A Fig). The spatial pattern of lymphocytes in the ILF resembles the architecture previously observed in Peyer's patches, which exhibit a core of B cells and a mantle of T cells [18]. Using smFISH, we further identified scattered localization of Rorc+Ccr6+ lymphoid tissue–induced (LTi) cells ([19]; S3B Fig), as well as radial zonation of dendritic cells and macrophages toward the periphery of the ILF (S3C Fig). We differentiated between macrophages and dendritic cells by the expression of *Itgax* and *C1qc* genes. While both intestinal dendritic cells and macrophages express *Itgax* (encoding CD11c) [20], *c1qc* is highly expressed in macrophages, but its expression is abrogated in mature dendritic cells [21].

The FAE segments exhibited elevated expression of M cell markers, including *Anxa5* (S4 Fig) that were higher in the FAET compared to FAEB and VB. Yet, M cells constituted only 8.8% ± 1.7% of all FAE cells (mean of 5 different smFISH images; S4C Fig). The FAET showed higher levels of genes previously shown to be elevated in FAE of Peyer's patches ([9]; S5A Fig)

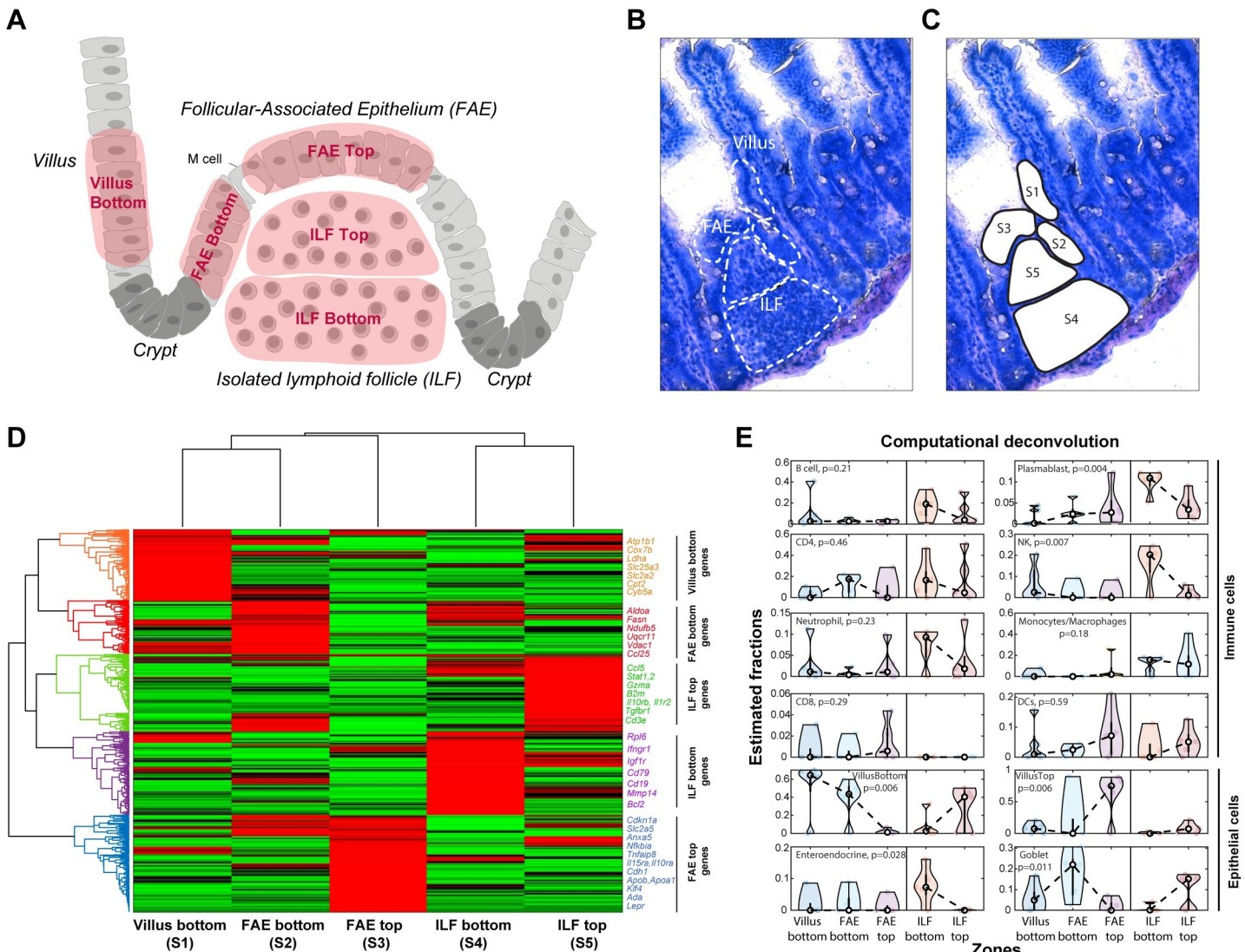

**Fig 1. LCM RNA-seq of FAE and ILF.** (**A**) An illustration of small intestinal region including FAE, ILF, adjacent villus, and crypts, showing the 5 dissected segments in red: VB (S1), FAEB (S2), FAET (S3), ILFB (S4), and ILFT (S5). (**B, C**) A bright field microscopy image (20× magnification) of FAE, ILF, and the adjacent villus before and after laser dissection of the 5 segments (S1–S5). (**D**) Clustergram of LCM RNA-seq data showing gene mean expression Z-score of the 5 segments (S1–S5). Selected genes with high expression are shown on the right side of the clustergram for each cluster, colored according to the cluster color. (**E**) Estimated fractions of distinct cell types, based on computational deconvolution of the LCMseq. White dots are medians, black boxes delineate the 25–75 percentiles. The data used to generate this figure can be found in Supporting information S1 and S2 Data. *P* values computed using Kruskal–Wallis tests. FAE, follicle-associated epithelium; FAEB, FAE bottom; FAET, FAE top; ILF, isolated lymphoid follicle; ILFB, ILF bottom; ILFT, ILF top; LCM RNA-seq, laser capture microdissection RNA sequencing; VB, villus bottom.

and lower levels of genes shown to be down-regulated in FAE of Peyer's patches ([8]; S5B and S5C Fig, Methods). Our deconvolution analysis indicated that the FAET segment was enriched in enterocytes with expression programs typical of the villus top (Fig 1E). We next turned to examine the zonated properties of the non-M cell epithelial cells, constituting 90% of this tissue compartment.

## Antimicrobial and absorption programs are down-regulated in FAE top

The mucosal layer that lines the intestinal epithelium is a critical barrier against bacteria. This barrier consists of a mucus layer that contains antimicrobial peptides. Previous in situ imaging

studies on FAE of the large Peyer's patches demonstrated a lower abundance of the mucus-secreting goblet cells [22,23], as well as a decline in enterocyte antimicrobial genes [6]. Our spatial analysis enabled global characterization of these and other zonated antimicrobial programs with high spatial resolution in FAE associated to the smaller ILFs.

Our LCMseq showed a depletion of goblet cells at the FAET and decreased levels of Muc2 in the FAE (Figs 1E and 2A). Using smFISH, we validated the lower abundance of Muc2+ goblet cells in the FAE with 3.6% ± 1.0% compared to 9.3% ± 1.1% in the adjacent villus (mean of 5 different smFISH images; Fig 2B). We also found a decline in the epithelial expression of *Pigr*, encoding the polymeric immunoglobulin receptor that facilitates transcytosis of IgA antibodies from the lamina propria to the lumen [6] (Fig 2A, 2C and 2I). *Nlrp6*, encoding a component of the microbial sensing inflammasome [24], also exhibited a steady decline from the VB to the FAET (Fig 2A, 2E and 2I). Unlike *Nlrp6*, *Muc2*, and *Pigr*, which decreased in expression in both the FAEB and FAET compared to the adjacent VB, we found that *Reg3g*, encoding a secreted antimicrobial peptide, decreased at the FAET but exhibited an increase in the FAEB, compared to the adjacent VB (Fig 2A, 2D and 2I). The decline in antimicrobial programs at the FAET may contribute to the required enrichment for bacteria at this sampling zone, whereas the compensatory higher expression at the FAE boundaries may help to prevent microbial access into the neighboring crypts. Notably, genes encoding ribosomal proteins exhibited similar spatial expression patterns to those of *Reg3g*, with a decrease in the FAET and a compensatory increase at the FAEB compared to an adjacent VB (S6A Fig). This increased expression of ribosomal proteins at the FAE boundaries could potentially facilitate the elevated translation of the secreted antimicrobial proteins in this zone.

Along with the reduction of antimicrobial programs, our analysis further revealed a steady decrease in key genes associated with nutrient absorption toward the FAET. These included the sodium-coupled transporter *Atp1b1*, the fatty acid binding proteins *Fabp1*, *Fabp*2, as well as different solute carrier genes, such as the glucose transporter *Slc2a2*, and the amino acid transporters *Slc7a7*, *Slc7a9* (Figs 2F–2I and S6B).

## FAE top enterocytes exhibit a partial villus tip program

Previous reconstruction of spatial gene expression patterns along the intestinal villus axis revealed a substantial up-regulation of dozens of genes in enterocytes at the villus tip [11]. These included cell adhesion genes such as *Cdh1*, encoding E-cadherin, apolipoproteins, stress-associated transcription factors such as *Jun*, *Fos*, and *Klf4*, as well as the purine metabolism immune-modulatory genes *Ada*, *Nt5e*, and *Slc28a2* (Fig 3A). We asked whether FAET enterocytes exhibit similar elevation of these genes. We found that some villus tip enterocyte genes, such as *Cdh1*, *Cdkn1a* (encoding the protein P21), and the apolipoprotein genes *Apoa1* and *Apoa4* were elevated at the FAET. In contrast, *Fos*, *Jun*, *Klf4*, and *Slc28a2* were not elevated at the FAET (Fig 3B–3K). The increase in Apoa proteins, which are essential components of chylomicrons used to transport absorbed lipids to the body, is surprising, given that the absorption machinery was largely down-regulated in the FAET zone (Figs 2F and S6B). This increase could be related to additional anti-inflammatory roles of apolipoproteins [25,26]. Our analysis therefore demonstrates that FAET enterocytes exhibit a partial induction of the expression program normally observed at the villus tip.

## FAE top telocytes are marked by Lepr and are distinct from villus tip telocytes

PDGFRα+ subepithelial telocytes are slender elongated cells that line the intestinal epithelium and have recently been shown to be important niche cells that shape zonated epithelial

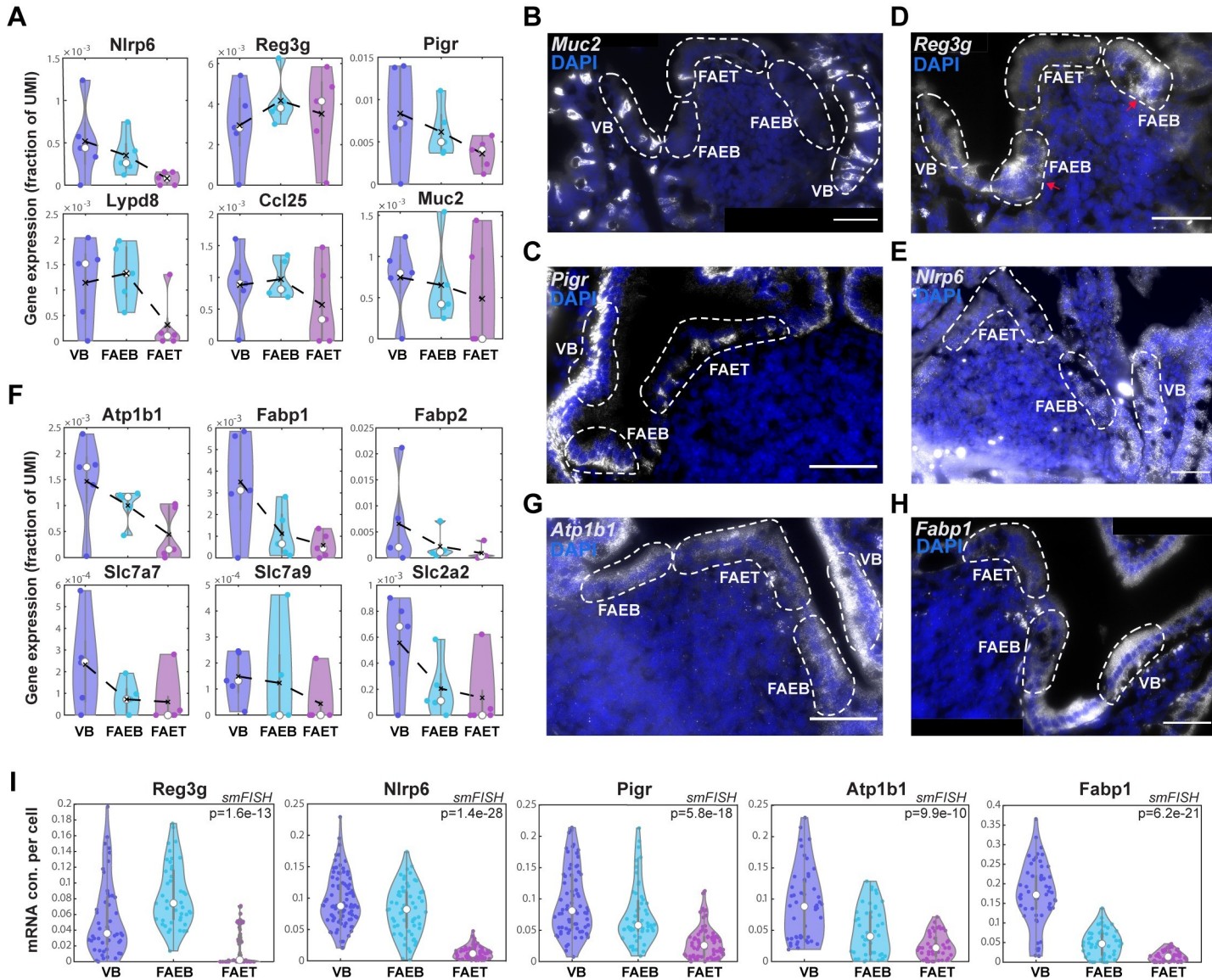

**Fig 2. Down-regulation in antimicrobial and absorption expression programs.** (**A, F**) Violin plots of LCMseq gene expression (fraction of UMI) showing down-regulated expression of antimicrobial genes (**A**) *Nlrp6*, *Reg3g*, *Pigr*, *Lypd8*, *Ccl25*, and *Muc2* and nutrient absorption genes (**F**) *Atp1b1*, *Fabp1*, *Fabp2*, *Slc7a7*, *Slc7a9*, and *Slc2a2* in FAET compared to FAEB and VB. Black dashed lines represent mean values; white dots represent median values of 5 repeats of at least 3 individual FAE from 2 mice. Gray boxes delineate the 25–75 percentiles. Kruskal–Wallis *p* > 0.05 for the genes in **A, F**. (**B-E, G-H**) Representative smFISH images showing decreased expression of *Muc2* (**B**), *Pigr* (**C**), *Reg3g* (**D**), *Nlrp6* (**E**), as well as *Atp1b1* (**G**) and *Fabp1* (**H**) in FAET compared to FAEB and VB. *Reg3g* reduced expression in FAET is compensated by higher expression in FAEB, marked with red arrows. White dashed lines delimit FAET, FAEB, and VB areas. DAPI staining for cell nucleus in blue. Scale bar: 50 µm. (**I**) Violin plots of dot quantifications of smFISH of *Reg3g*, *Nlrp6*, and *Pigr*, as well as *Atp1b1* and *Fabp1* (analysis performed over 4 mice with 3–5 individual ILF/FAEs per mouse). White dots are median values; gray boxes delineate the 25–75 percentiles. The data used to generate this figure can be found in Supporting information S2 and S5 Data. FAE, follicle-associated epithelium; FAEB, FAE bottom; FAET, FAE top; ILF, isolated lymphoid follicle; ILFB, ILF bottom; ILFT, ILF top; smFISH, single-molecule fluorescence in situ hybridization; UMI, unique molecular identifier; VB, villus bottom.

expression programs [27,28]. Villus tip telocytes, marked by the gene *Lgr5*, are regulators of the villus tip enterocyte program [29]. We argued that our LCM RNA-seq of the epithelial segments may also include subepithelial telocytes, in addition to epithelial cells, due to their adjacency to the epithelial layer. To identify potential telocyte gene expression, we extracted 1,765 genes that were previously shown to be substantially more highly expressed in telocytes compared to enterocytes (Methods). Among these potential telocyte marker genes, we found elevated expression at

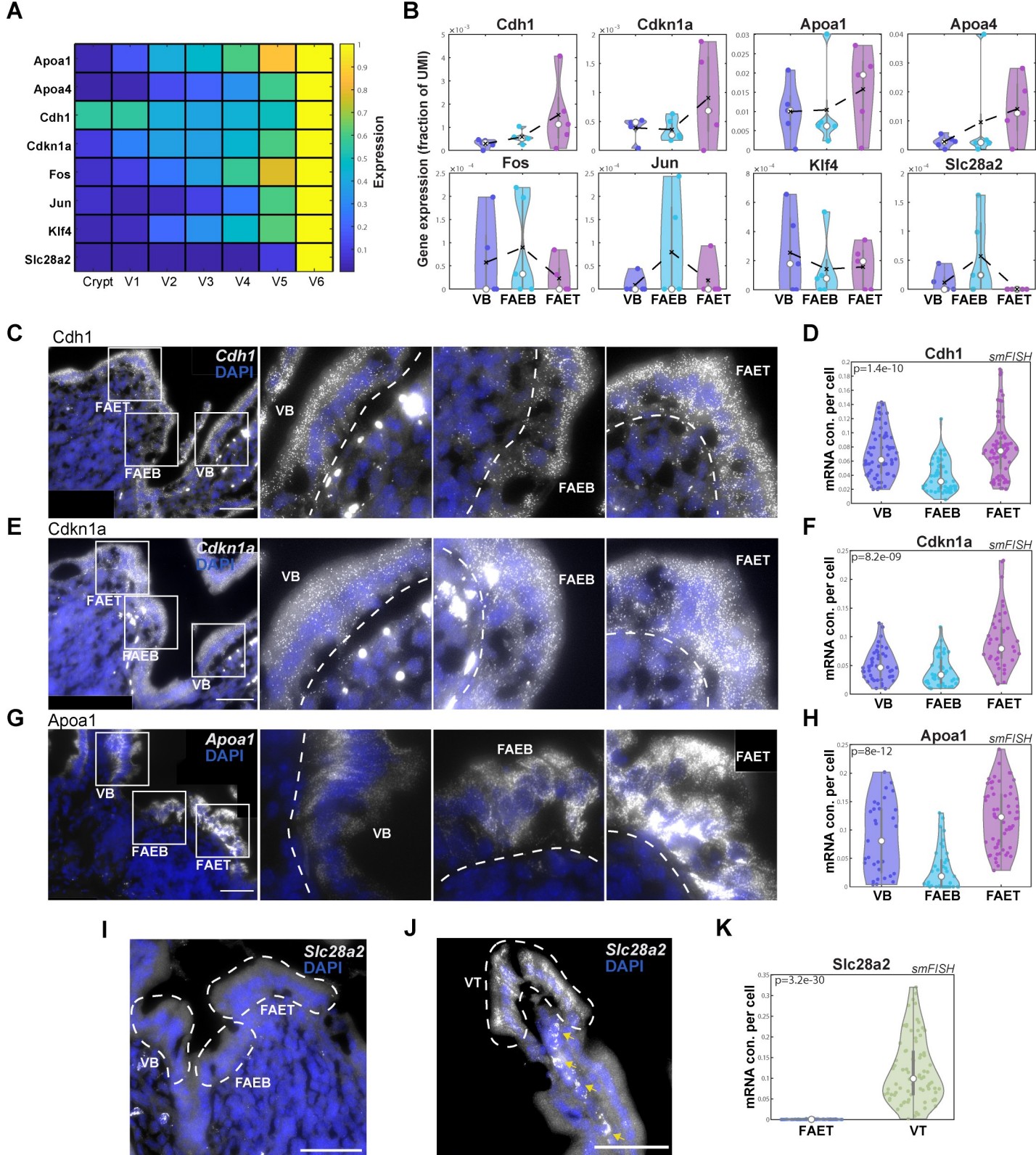

**Fig 3. FAET enterocytes exhibit a partial VT program.** (**A**) Heatmap of epithelial gene expression along the crypt-villus axis (V1-VB and V6-villus top), showing up-regulated expression of tip program genes (data from [11]). (**B**) Violin plots of gene expression (fraction of UMIs) showing increased expression of *Cdh1*, *Cdkn1a*,

*Apoa1*, and *Apoa4*, but not *Fos*, *Jun*, *Klf4*, and *Slc28a2* at the FAET. Black dashed lines represent mean values; white dots represent median values of 5 repeats of at least 3 individual FAE from 2 mice; gray boxes delineate the 25–75 percentiles. Kruskal–Wallis *p* > 0.05 for these LCM measurements. (**C, E, G**) Representative smFISH images showing up-regulated expression of *Cdh1* (**C**), *Cdkn1a* (**E**), and *Apoa1* (**G**) in the FAET compared to FAEB and VB. In each panel, left image shows the ILF/FAE, followed by magnifications of the labeled white-boxed regions. Scale bar is 50 μm. DAPI staining of cell nucleus in blue. White dashed lines delimit FAET, FAEB, and VB areas. (**D, F, H**) Violin plots of dot quantifications of smFISH experiments. *P* values computed using Kruskal–Wallis tests. (**I, J**) Representative smFISH images showing no expression of *Slc28a2* in FAET, FAEB, and VB, and high expression at the VT. Yellow arrows point at autofluorescent blobs. (**K**) Violin plot of smFISH dot quantification of *Slc28a2* expression in FAET compared to VT. The smFISH validations (**D, F, H, K**) showing the concentration (con.) of dots (mRNA molecules) per cell area (analysis performed over 4 mice with 3–5 individual ILF/FAEs per mouse). In all violin plots, white dots represent median values; gray boxes delineate the 25–75 percentiles. The data used to generate this figure can be found in Supporting information S2 and S5 Data. FAE, follicle-associated epithelium; FAEB, FAE bottom; FAET, FAE top; ILF, isolated lymphoid follicle; ILFB, ILF bottom; ILFT, ILF top; smFISH, single-molecule fluorescence in situ hybridization; UMI, unique molecular identifier; VB, villus bottom; VT, villus tip.

the FAET of *Lepr*, encoding the leptin receptor (Fig 4A and 4B). *Lepr* was previously shown to be elevated in crypt mesenchymal cells in the colon [30]. Using smFISH combined with immunostaining for PDGFRα, we validated that FAET telocytes express significantly higher levels of *Lepr* compared to the FAEB and adjacent VB (Fig 4C and 4F). In contrast, *Lgr5* was highly expressed in villus tip telocytes (as well as in the crypt epithelial stem cells) but was undetectable in the subepithelial layers of the FAE (Fig 4C–4E). Our analysis therefore identifies *Lepr+* subepithelial telocytes distinct from the *Lgr5+* telocytes at the villus tip.

## Discussion

ILFs, along with Peyer's patches, are critical sites of intestinal immune surveillance, yet their scarcity and small sizes render them hard to isolate. Our analysis provides a comprehensive spatial atlas of gene expression of ILFs and their associated epithelium. We found that FAE enterocytes exhibit a gene expression signature that is distinct from both enterocytes at the VB and at the villus tip. FAET enterocytes express lower levels of antimicrobial genes, as well as lower levels of the nutrient absorption genes. These features seem to be tuned toward maintaining a microenvironment that is optimal for efficient sampling of bacterial antigens by M cells and immune cells, rather than nutrient absorption.

What are the factors that could elicit the distinct gene expression programs at the FAET? FAE enterocytes are localized at around the same physical distance from the crypt as VB enterocytes. Moreover, previous work has shown that FAE enterocytes are continuously migrating and shedding from the FAE tip, similarly to villus enterocytes [31]. Yet, the FAE operates under a unique microenvironment compared to the VB. Luminal bacterial concentrations at the FAE tip should be higher due to the thinner mucous layer and reduced secretion of antimicrobial peptides, and, therefore, more similar to the luminal microenvironment at the villus tip [11]. We found that, unlike villus tip enterocytes, FAET enterocytes are in contact with *Lepr+* telocytes, which could provide different niche signals than their villus tip *Lgr5+* telocytes counterparts. Indeed, the expression of the purine metabolism immune-modulatory genes *Ada*, *Nt5e*, and *Slc28a2* at the villus tip seems to be controlled by *Lgr5+* telocytes [29], potentially explaining their reduced expression at the FAET. Our study forms the basis for the future exploration of the regulatory molecules that shape FAE zonation. It will be interesting to expand our study to CPs and colonic ILFs, as well as to the characterization of ILFs in perturbed states such as germ-free mice and in models of inflammatory diseases.

## Methods

### Animal experiments

All animal studies were approved by the Institutional Animal Care and Use Committee (IACUC, 08391020–3) of the Weizmann Institute of Science. C57bl6 male mice at the age of

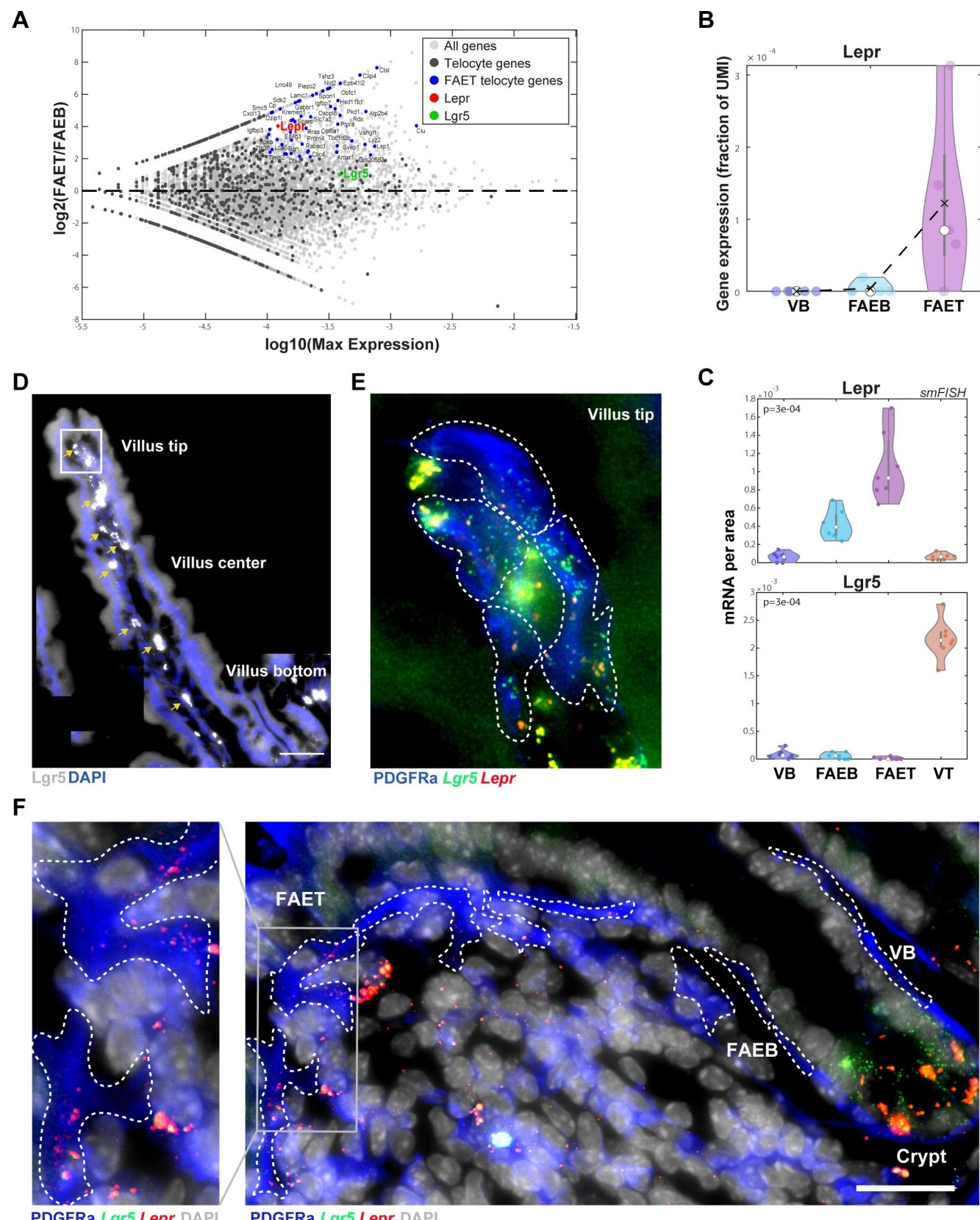

**Fig 4. Lepr+ telocytes are zonated to FAET.** (**A**) MA plot showing log2 of the expression ratios between FAET and FAEB vs. log10 of the max-expression. Gray dots: all genes; dark gray dots: telocyte genes; blue dots: highly expressed telocyte genes up-regulated in FAET. *Lepr* gene marked in red and *Lgr5* gene in green (telocyte expression curated from previous scRNAseq papers; see Methods). (**B**) Violin plot showing high expression of *Lepr* (as a fraction of UMIs) in FAET. Black dashed lines represent mean values; white dots represent median values of 5 repeats of at least 3 individual FAE from 2 mice; gray boxes delineate the 25–75 percentiles. Kruskal–Wallis $p > 0.05$. (**C**) Violin plots of quantification of smFISH experiments for *Lepr* and *Lgr5* showing mRNA concentrations (con.) per zone in smFISH images (7 repeats of different zones from 4 mice). $P$ values computed using Kruskal–Wallis tests. White dots represent median values; gray boxes delineate the 25–75 percentiles. (**D-F**) smFISH validations of *Lgr5* and *Lepr* expression in villus tip and FAET. (**D**) smFISH image showing *Lgr5* expression in gray of a full-length villus (tip, center and bottom). Yellow arrows point at autofluorescent blobs. Scale bar: 50 μm. (**E**) Blowup of villus tip showing *Lgr5* expression (green) but not *Lepr* expression (red) in PDGFRα telocytes. (**F**) smFISH image showing *Lepr* expression in gray in FAE. Scale bar: 25 μm. Blowup showing *Lepr* expression (red) in PDGFRα telocytes (blue) in FAET but not in FAEB and VB. *Lgr5* expression is absent from FAET telocytes and observed in the crypt stem cells. White dashed lines delimit PDGFRα telocytes. The data used to generate this figure can be found in Supporting information S2 and S5 Data. FAE, follicle-associated epithelium; FAEB, FAE bottom; FAET, FAE top; scRNAseq, single-cell RNA sequencing; smFISH, single-molecule fluorescence in situ hybridization; UMI, unique molecular identifier; VB, villus bottom.

12 to 15 weeks (Harlan Laboratories) were kept in SPF conditions and fed with regular chow ad libitum. LCM experiments were conducted on the Jejunum segments extracted from 2 WT mice (see details below). Tissues were transected on top of Wattman paper soaked in cold PBS and then were immediately embedded without fixation in OCT (Scigen, 4586) on dry ice. Frozen blocks were transferred to −80˚C until use. Each area is a pool of at least 3 individual FAE/ILF dissected from 10 serial sections, 10 μm each, with a total of at least 50,000 μm². For smFISH experiments, jejunum tissues were harvested from 4 WT mice and fixed in 4% formaldehyde (J.T. Baker, JT2106) for 3 h, incubated overnight with 30% sucrose in 4% formaldehyde, then embedded in OCT in the form of Swiss rolls and were kept at −80˚C until use.

## Laser capture microdissection (LCM)

LCM experiments were performed as was previously described [11] with some modifications. Briefly, 10 serial sections of 10 μm thickness were attached on polyethylene-naphthalate membrane-coated 518 glass slides (Zeiss, 415190-9081-000), air dried for 20 s at room temperature, washed in 70% ethanol for 25 s, incubated in water for 25 s (Sigma-Aldrich, W4502), stained with HistoGene Staining Solution for 20 s (Thermo Fisher Scientific, KIT0401), and washed again in water for 25 s. Stained sections were dehydrated with subsequent 25 s incubations in 70%, 95%, and 100% EtOH and air dried for 60 s before microdissection. The experiments were performed using UV laser-based unit (PALM-Microbeam, Zeiss) and a bright field imaging microscope (Observer.Z1, Zeiss). PALM ×40 lenses were used to catapult and collect ILF and FAE tissue segments into 0.2 ml adhesive cap tubes (Zeiss, 415190-9191-000). ILFs were identified histologically as approximately 200 μm width aggregates of immune and stromal cells in association to a dome-like epithelial layer. In total, 25 LCM samples were collected for RNA sequencing, containing 5 repeats from 2 mice (rep1 to rep2 for mouse#1 and rep3 to rep5 from mouse#2). Each repeat included a set of 5 areas: FAET, FAEB, VB, ILFT, and ILFB of at least 3 independent ILF/FAE, with a total area of 50,000 μm² per segment. Pooled sections were resuspended in 7 μl lysis buffer: RLT lysis buffer (Qiagene, 1015762) supplemented with 0.04 M DTT, immediately transferred to dry ice and stored at −80˚C until the preparation of RNA libraries.

## LCM RNA-seq

Sequencing of the LCM-extracted segments was performed using the mcSCRB-seq protocol [15]. This protocol incorporates UMIs, enabling corrections of potential imbalanced PCR amplifications of transcripts. LCM samples were washed from lysis buffer by mixing with an equal portion of SPRI bead solution (Beckman Coulter, A63881), incubated 5 min R/T and washed twice with 100% EtOH under magnetic field. RNA captured on bead was elucidated

using 10 μl of RT reaction mixture (Thermo Fisher) (1× Maxima H Buffer, 1 mM dNTPs, 2 μM TSO* E5V6NEXT, 7.5% PEG8000, 20U Maxima H enzyme, 1 μl barcoded RT primer) and was taken to RT reaction (42°C, 90 min and inactivation in 80°C, 10 min). The subsequent steps were applied as previously reported in the of cDNA library preparation protocol for mcSCRB-seq with the following modifications. The cDNA was amplified with 15 to 18 cycles, depending on the cDNA concentration indicated by qRT-PCR quality control. Then, 0.6 ng of the amplified cDNA was converted into the sequencing library with the Nextera XT DNA Library Preparation Kit (Illumina, FC-131-1024), according to the protocol supplied. Quality control of the resulting libraries was performed with a High Sensitivity DNA ScreenTape Analysis system (Agilent Technologies, 5067–5584). Library concentration was determined using the NEBNext Library Quant Kit (Illumina, E7630). Libraries were diluted to a final concentration of 2.8 pM in HT1 buffer (supplemented with the kit) and loaded on 75-cycle high-output flow cells (Illumina, FC-404-2005) and sequenced on a NextSeq 550 (Illumina). Raw and processed sequencing data are available in the GenBank GEO database NCBI (GSE168483).

## Bioinformatic analysis

Illumina output sequencing raw files were converted to FASTQ files using bcl2fastq package. To obtain the UMI counts, FASTQ reads were aligned to the mouse reference genome (GRCm38.84) using zUMI package [32] and STAR with the following parameters: RD1 16 bp, RD2 66 bp with a barcode (i7) length of 8 bp. For UMI table, see S2 Data. Subsequent analysis was done with Matlab R2018b. UMI table was filtered to retain only protein coding genes, based on GRcm38.84 ensembl acquired via BioMart using biomaRt R package version 2.44.1 (S2 Data). Mitochondrial genes were further removed, and the remaining UMI counts for each sample were normalized to the sum of UMI counts of all genes that individually took up less than 5% of the UMI count sum in any of the samples. Mean expression was calculated over all samples from the same zone (S2 Data). Gene expression was calculated as fraction of UMI counts over the sum of UMIs (S2 Data) and was presented in violin plots. Clustering was performed using the Clustergram function in Matlab with correlation distance over all genes with maximal mean expression above $10^{-4}$. Gene set enrichment analysis was done using MSigDB_Hallmark_2020 database of Enrichr [33,34]. The list of genes in each set is specified in S3 Data. For extracting telocyte genes, we used previous datasets of cell type–specific gene expression in the mouse small intestine and compared the mean expression over villus tip and crypt telocytes [29] to the mean expression over zonated crypt-villus populations [11], goblet cells, and enteroendocrine cells [8]. Telocyte genes were defined as those with mean expression above $10^{-5}$ in telocytes and 4-fold higher expression in telocytes compared to epithelial cells. M cell marker genes (S3G Fig) were obtained from previous analyzed data [8]. To identify T cell and B cell markers, we analyzed a previous scRNAseq dataset [35]. We calculated the maximal mean expression among cells annotated as CD4 or CD8 and compared it to the mean expression of B cells. We considered genes with expression above $5*10^{-4}$ in the corresponding cell type and extracted the 20 genes with the highest expression ratio between the 2 cell types. Gene marker lists generated of this analysis are presented in S6 Data. Kruskal–Wallis tests were used to assess statistical significance as $p$-value $< 0.05$.

The list of epithelial genes previously shown to be up-regulated in FAE of Peyer's patches using microarray measurements ([9]; S5A Fig) was generously provided by Hiroshi Saito and Koji Hase. Epithelial genes up- or down-regulated in FAE of Peyer's patches based on Haber and colleagues ([8]; S5B and S5C Fig) were extracted as follows: FAE cells were extracted from the FAE scRNAseq dataset and defined as cells annotated as "Enteroproximal" with UMI sum-normalized expression of the villus top gene Apoa1 above $10^{-3}$. Villus cells were extracted

from the Regional scRNAseq dataset and defined as jejunum cells with UMI sum-normalized expression above $10^{-3}$. The 2 datasets were normalized over the set of intersecting genes, after removal of epithelial secretory cell genes, defined as genes with expression in epithelial secretory cells that is larger than $10^{-5}$ and larger than 2-fold the expression in enterocytes. To this end, epithelial secretory cell gene expression was defined as the maximal mean expression among Paneth cells, tuft cells goblet cells, and enteroendocrine cells, extracted from Haber and colleagues [8]. The enterocyte expression was defined as the maximal expression across all crypt-villus zones, based on Moor and colleagues [11]. Differential gene expression among the sets was performed using Wilcoxon rank sum tests using Benjamini–Hochberg multiple hypotheses correction. FAE up/down-regulated genes were defined as genes with mean expression above $10^{-5}$, 2-fold or higher/lower expression, respectively, in FAE cells compared to villus cells and q-value below 0.1.

Computational deconvolution was performed using CIBERSORTx [17]. CIBERSORTx was run with the "Impute Cell Fractions" analysis module using custom analysis mode with default settings. Mixture file included the LCMseq samples; signature files were extracted based on single-cell atlases of the mouse intestine (S1 Data). VB and top enterocytes were defined as the averages of the lower or upper 3 villus zones in Moor and colleagues [11], respectively. Goblet cells and enteroendocrine cells signatures were extracted based on Haber and colleagues [8]. Immune cell signatures were extracted based on Biton and colleagues [35]. All dendritic cell subtypes were averaged and coarse-grained into one group, as were monocytes and macrophages. Paneth cells, crypt cells, and tuft cells were not included in the input signatures due to their expected lower abundance in the tissue compartments analyzed. Both mixture and signature tables were renormalized to the sum of the overlapping genes, genes for which the maximal expression among the mixture samples was either 10-fold higher or 10-fold lower than the maximal expression among signature samples were removed and data were subsequently renormalized so the that each sample sums up to $10^6$.

## Single-molecule FISH (smFISH)

The smFISH experiments were conducted as was previously described [13] with some modifications. Briefly, 8 μm thick sections of fixed jejunum were sectioned and placed onto poly L-lysine (Sigma-Aldrich, P8920) coated coverslips, then fixed again in 4% FA in PBS for 15 min R/T followed by 70% ethanol dehydration for 2 h in 4°C. Tissues were treated for 10 min R/T with proteinase K (10 μg/ml Ambion AM2546) and washed twice with 2× SSC (Ambion AM9765), then incubated in wash buffer (20% Formamide Ambion AM9342, 2× SSC) for 10 min R/T. Next, the tissues were mounted with the hybridization buffer (10% Dextran sulfate Sigma D8906, 20% Formamide, 1 mg/ml *E. coli* tRNA, Sigma R1753, 2× SSC, 0.02% BSA, Ambion AM2616, 2 mM Vanadyl-ribonucleoside complex, NEB S1402S) mixed with 1:3,000 dilution of specific probe libraries and were transferred to an overnight incubation at 30°C. Probe libraries were designed using the Stellaris FISH Probe Designer Software (Biosearch Technologies, Petaluma, CA). For probe library list, see S4 Data. After overnight incubation, the tissues were washed with wash buffer supplemented with 50 ng/ml DAPI (Sigma-Aldrich, D9542) for 30 min at 30°C and washed with GLOX buffer (0.4% Glucose, 1% Tris, 10% SSC). For the detection of telocytes, goat anti-PDGFRα primary antibody (AF1062 R&D Systems) (8 μg/μl) was added to the smFISH hybridization buffer and Alexa Fluor 488 conjugated donkey anti-goat as secondary antibody (Jackson Laboratories, 705-545-147, 1:400) in GLOX buffer for 20 min after staining with DAPI.

Imaging was performed on Nikon eclipse Ti2 inverted fluorescence microscopes equipped with 100× oil-immersion objectives and a Photometrics Prime 95B 25MM EMCCD camera.

All images were taken using ×100 magnifications; therefore, several fields of view were stitched together to cover the whole ILF/FAE area using Fiji software. Images were processed with a Laplacian of Gaussian filter [13]. Quantification of smFISH was done using ImageM [14] over 4 mice using 3 to 5 imaged tissues from each. Images were segmented, and fluorescent dots were counted and divided by the segmented cell volume to obtain the mRNA concentration per cell. Segmentation of ILFT and ILFB was done using a border line in the middle of the ILF that was determined based on DAPI staining of cell nucleus and blinded to the smFISH channel. Specifically, segmentation and dot counting of telocytes was done using Fiji based on the immune-fluorescent staining of the surface expressed PDGFRα. Region of interest (ROI) was manually segmented, and dots were counted per area of 7 different images from 4 mice. Kruskal–Wallis tests were used to assess statistical significance as $p$-value < 0.05.

## Supporting information

**S1 Fig. Gene set enrichment. Related to Fig 1**. (**A-E**) Dot plots generated by Enrichr (see Methods) showing normalized combined score (with q-value less than 0.1) of up-regulated gene sets for VB, FAEB, FAET, ILFB, and ILFT. The size of each red dot is in accordance with the number of genes in each set. The data used to generate this figure can be found in Supporting information S3 Data. FAEB, FAE bottom; FAET, FAE top; ILFB, ILF bottom; ILFT, ILF top; VB, villus bottom.
(TIF)

**S2 Fig. Zonation of B and T cells in ILF. Related to Fig 1.** (**A-C, G-H**) smFISH validations showing increased expression of *Cd19* (**A**) at the ILFB and *Cd3e* (**B**), *Gzma* (**C**), *Cd4* (**G**), *Cd8* (**H**) at the ILFT. White dashed lines delimit segment ILF areas, and a border line in the middle separates ILFT and ILFB. Red arrows highlight cells with elevated expression of the respective genes. DAPI staining for cell nucleus in blue. Scale bar: 50 μm. (**D-F**) Violin plots of dot quantifications of smFISH signals of *Cd19*, *Cd3e*, and *Gzma*, showing the concentration (con.) of dots (mRNA molecules) per cell area (3–5 individual ILF per mouse for 4 mice). Blowup in (**F**) highlights the majority of the cells with lower expression levels, demonstrating the increase in median *Gzma* levels at the ILFT. (**I**) Violin plots showing max-normalization of B and T cell signature gene expression (Methods). B cell markers are significantly zonated to the ILFB, whereas T cell markers do not exhibit significant bias to the ILFB. White dots are median values; gray boxes delineate the 25–75 percentiles, $p$-values computed using Kruskal–Wallis tests. The data used to generate this figure can be found in Supporting information S2, S5, and S6 Data. FAEB, FAE bottom; FAET, FAE top; ILF, isolated lymphoid follicle; ILFB, ILF bottom; ILFT, ILF top; smFISH, single-molecule fluorescence in situ hybridization.
(TIF)

**S3 Fig. Immune cell subsets in ILFs. Related to Fig 1**. (**A-C**) smFISH images and blowups showing *Gzma* expressing cytotoxic T cells at the ILFT, infiltrating to the FAET (**A**), as well as *Ctla4*+ Tregs (**A**) and *Rorc+Ccr6*+ LTi cells (**B**) that are scattered throughout the ILFs. Red arrows highlight representative cells. (**C**) Dendritic cells (Itgax+C1qc− cells) and macrophages (Mϕ, Itgax+C1qc+ cells) are radially zonated toward the periphery of the ILF. Scale bar-50 μm. DC, dendritic cell; FAEB, FAE bottom; FAET, FAE top; ILF, isolated lymphoid follicle; ILFB, ILF bottom; ILFT, ILF top; LTi, lymphoid tissue–induced; smFISH, single-molecule fluorescence in situ hybridization; Treg, regulatory T cell.
(TIF)

**S4 Fig. Zonation of M cells in the FAE. Related to Fig 1** (**A**) Violin plots for the max-normalized expression of M cell signature genes (Methods) showing an up-regulation expression of

M cell genes in FAET compared to FAEB and VB. (**B**) Violin plot showing up-regulation of Anxa5 expression in FAET compared to FAEB and VB. Five FAE zones from 2 mice. In **A, B,** white dots are median values; gray boxes delineate the 25–75 percentiles, *p*-values computed using Kruskal–Wallis tests. $P > 0.05$ for Anxa5 (**B**). (**C**) A smFISH image showing *Anxa5* expression in the FAET. White dashed lines delimit FAET and FAEB areas, and red arrows mark cells with higher expression levels of *Anxa5*. Scale bar: 50 μm. The data used to generate this figure can be found in Supporting information S2 and S6 Data. FAE, follicle-associated epithelium; FAEB, FAE bottom; FAET, FAE top; smFISH, single-molecule fluorescence in situ hybridization; UMI, unique molecular identifier; VB, villus bottom.
(TIF)

**S5 Fig. Representation of genes previously shown to be differentially expressed in FAE in Peyer's patches. Related to Fig 1.** (**A**) Log10 of the average expression in each of the epithelial zones for RefSeq genes previously shown to be differentially expressed in FAE of Peyer's patches [9]. (B, C) Log10 of the average expression of genes that are differentially expressed between FAE enterocytes and villus enterocytes in the jejunum ([8]; Methods). Bottom plots show mean of log10 expression of the gene sets; patches are standard errors of the means. The data used to generate this figure can be found in S1 Data. FAE, follicle-associated epithelium; FAEB, FAE bottom; FAET, FAE top; VB, villus bottom.
(TIF)

**S6 Fig. Down-regulated expression of ribosomal proteins and solute carriers in FAE top. Related to Fig 2.** (**A, B**) Violin plots showing down-regulated max-normalized expression of ribosomal proteins (**A**) and solute carriers (**B**) in FAET compared to FAEB and VB. *P* values of Kruskal–Wallis tests are presented. White dots represent median values. The data used to generate this figure can be found in Supporting information S2 and S6 Data. FAE, follicle-associated epithelium; FAEB, FAE bottom; FAET, FAE top; VB, villus bottom.
(TIF)

**S1 Data. CIBERSORTx_results.** First tab presents the results of the computational deconvolution; second tab presents the input signature table; third tab presents the input mixture table.
(XLSX)

**S2 Data. UMI table, filtered UMI table, and mean expression table.** (**Tab1**) UMI table of 24,090 genes expressed in S1 (VB), S2 (FAEB), S3 (FAET), S4 (ILFB), and S5 (ILFT) of 5 experimental repeats obtained from 2 mice (Mouse#1 and #2). (**Tab2**) UMI table of only protein coding genes (16,312 genes). (**Tab3**) Mean expression and standard error of the 5 experimental repeats. FAEB, FAE bottom; FAET, FAE top; ILFB, ILF bottom; ILFT, ILF top; UMI, unique molecular identifier; VB, villus bottom.
(XLSX)

**S3 Data. Output of enriched set of genes from Enrichr analysis.** Five tabs for each segment: S1 (VB), S2 (FAEB), S3 (FAET), S4 (ILFB), and S5 (ILFT) showing an output of Enrichr analysis (Methods) for up-regulated gene sets (Terms), including *p*-values, adjusted *p*-values, odds ratio, and combined score. The genes in each set are specified in the last column. FAEB, FAE bottom; FAET, FAE top; ILFB, ILF bottom; ILFT, ILF top; VB, villus bottom.
(XLSX)

**S4 Data. Probe list.** A list of probe libraries used for smFISH experiments in this study.
(XLSX)

**S5 Data. Raw data of smFISH validations.** Thirteen smFISH single-cell measurements, one gene per tab.
(XLSX)

**S6 Data. Gene markers.** Lists of gene markers of Ribosomes, solute carriers, M cells, B cells, and T cells.
(XLSX)

## Acknowledgments

We thank Moshe Biton for valuable discussions.

## Author Contributions

**Conceptualization:** Noam Cohen, Shalev Itzkovitz.

**Data curation:** Noam Cohen, Hassan Massalha, Milena Rozenberg, Keren Bahar Halpern.

**Formal analysis:** Shani Ben-Moshe, Adi Egozi, Shalev Itzkovitz.

**Investigation:** Noam Cohen.

**Methodology:** Noam Cohen.

**Software:** Adi Egozi, Shalev Itzkovitz.

**Supervision:** Shalev Itzkovitz.

**Writing – original draft:** Noam Cohen, Shalev Itzkovitz.

**Writing – review & editing:** Shalev Itzkovitz.

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
