## [Editor Report · Decision Letter 0]

30 Mar 2021

Dear Shalev, 

Thank you for submitting your manuscript entitled "Spatial expression programs of the intestinal follicle-associated epithelium" for consideration as a Short Report by PLOS Biology.

Your manuscript has now been evaluated by the PLOS Biology editorial staff as well as by an academic editor with relevant expertise and I am writing to let you know that we would like to send your submission out for external peer review.

Please re-submit your manuscript within two working days, i.e. by Apr 01 2021 11:59PM.

Kind regards,

Ines

--

Ines Alvarez-Garcia, PhD,

Senior Editor

PLOS Biology

---

## [Decision Letter · Decision Letter 1]

24 May 2021

Dear Shalev,

Thank you very much for submitting your manuscript entitled "Spatial expression programs of the intestinal follicle-associated epithelium" for consideration as a Short Report at PLOS Biology. Thank you also for your patience as we completed our editorial process, and please accept my apologies for the delay in providing you with our decision. Your manuscript has been evaluated by the PLOS Biology editors, an Academic Editor with relevant expertise, and by two independent reviewers.

As you will see, the reviewers find the results novel and interesting, but they also raise several issues that will need to be addressed. Reviewer 1 would like to see experiments addressing the function of the Lepr+ telocyte population, and changes in the structure in germ free animals or disease models or humans. However, after discussing these experiments with the academic editor, we do not think they are necessary for a Short Report. Nevertheless, we do agree with this reviewer that you should do more analysis of the immune cell types present as we consider this quite key for the spatial organisation of the structure. In addition, we would expect you to address the following points in a revision:

Reviewer 1:

1. You should compare your data to the studies analysing the transcriptome of FAE in Peyer's Patches (manuscript references 8-10).

2. Which T cell types are present?

3. Where are the dendritic cells and the LTis?

Reviewer 2:

1. The point regarding the low number of samples and statistical analyses should be addressed, particularly for the smFISH which seems to be only from two mice, but is subjected to statistical analysis. Are these statistics valid? Standard convention is to have three biological replicates prior to statistical treatment of the data.

2. Please avoid jargon in the manuscript to make it accessible to a broad audience.

Minor:

In Supplementary Figure 3E, CD3a is increased at ILFT (not at ILFB) - the figure legend seems incorrect while the main text is correct. In addition, on Supplementary Fig. 3E the medians for Gmza for ILFT and ILFB seem very similar according to the violin plot.

In light of the reviews (attached below), we will not be able to accept the current version of the manuscript, but we would welcome re-submission of a revised version that takes into account the reviewers' comments. We cannot make any decision about publication until we have seen the revised manuscript and your response to the reviewers' comments. Your revised manuscript is also likely to be sent for further evaluation by the reviewers.

We expect to receive your revised manuscript within 3 months. 

**IMPORTANT - SUBMITTING YOUR REVISION**

3. Resubmission Checklist

a) *Published Peer Review*

b) *PLOS Data Policy*

Please provide the data underlying the following figures, and make sure you mention in the corresponding figure legends WHERE THE DATA CAN BE FOUND:

Fig. 2A, F, I; Fig. 3B, D, F, H, K; Fig. 4A-C; Fig. S1A-E; Fig. S2A-D; Fig. S3D-H and Fig. S4A, B

c) *Ethics Statement*

Thank you for providing the Ethics Statement. Please include the license number.

Best wishes,

Ines

--

Ines Alvarez-Garcia, PhD

Senior Editor

PLOS Biology

Reviewers’ comments

Rev. 1:

In their manuscript Cohen et al, made a short and concise description of the expression profiles of isolated lymphoid follicles and their associated epithelium. The authors use a combination of laser capture micro dissection, RNAsequencing and smFISH to identify areas with unique expression profiles. The authors also describe the identification of Lepr+ sub-epithelial telocytes.

The study shows that according to their transcriptome profile ILFs could be divided into different compartments, which is a novel and interesting information for mucosal immunologists and epithelial biologists, but provides limited information on the biological relevance of such finding. There is no further insight into the function of ILF, FAE or the newly described Lepr+ sub-epithelial telocytes.

The authors should compare their data to the studies analysing the transcritptome of FAE in Peyer's Patches (manuscript references 8-10).

The characterisation of the immune cells present in the ILF compartments should be more detailed.

Which T cell types are present?

Where are the dendritic cells and the LTis?

For example, the data on figure 1 suggest that regulatory T cells are enriched on ILFB has Tregs associated genes Foxp3 and Ctla4 are enriched in this area.

On this subject, on Fig Sup 3E, CD3a is increased at ILFT (not at ILFB) - the figure legend is incorrect while main text is correct. In addition, on Fig Sup 3E, according to the violin plot, the medians for Gmza for ILFT and ILFB seem very similar.

Using their RNAseq data and SmFISH authors identified Lepr+ telocytes. What's the function of these telocytes?

As the authors mention it would be interesting to characterise IFLs in disease models and germ free animals. In addition, those this compartmentalisation also occurs in human ILFs?

The manuscript describes the bioinformatic analysis of the data generated - I don't have the expertise to comment on that.

Rev. 2:

In this study, the author use Laser-capture dissection and RNA-sequencing of a few initial samples to identify expression programs in different areas of isolated lymphoid follicles (ILF) in the mouse epithelium. They use the genes identified in this approach to characterize spatial expression using extensive single molecule FISH. As this small structure has not been investigated on a molecular level with respect to its cell type compositions, the author's analyses are certainly helpful to better understand it and its relevance for immune surveillance in the gut. However, I find the manuscript and the data not advanced enough to be really helpful for a larger readership at this point.

1) I think technically the LCM-seq is fine, but the experimental design is unclear and the number of samples is too low. The number of replicates is not even mentioned in the main text, the figure is a very ad hoc description of the central aspect of the manuscript and statistical analyses that take into account biological replicates and the repeated measures is also totally mssing..

2) The structure of how these initial results are followed-up with smFISH is very unclear, at least to me as a non-specialist in intestine cell types. Some genes are picked rather ad hoc from the list, subjected to smFISH and then interpreted. I can't detect a transparent structure in the manuscript which questions are addressed and how they are answered. A potentially more transparent way to tackle this would be to deconvolute the LCM-seq data to predict certain cell-types and then confirm them with smFISH… but this is certainly not the only possibility.

3) If the paper should be accessible to a wider readership beyond the immediate field the jargon and abbreviations need to be reduced and better introduced.

---

## [Decision Letter · Decision Letter 2]

23 Sep 2021

Dear Shalev,

Thank you for submitting your revised Short Report entitled "Spatial expression programs of the intestinal follicle-associated epithelium" for publication in PLOS Biology. I have now obtained advice from the original reviewers and have discussed their comments with the Academic Editor. 

Based on the reviews, we will probably accept this manuscript for publication, provided you satisfactorily address the remaining policy-related requests.

In addition, we would like you to consider a couple of alternatives to improve the title:

"Spatial gene expression maps of the intestinal lymphoid follicle and associated epithelium identify zonated expression programs"

or

"Spatial gene expression maps of the intestinal lymphoid follicle and associated epithelium identify zonated functional programs"

Please take this last chance to review your reference list to ensure that it is complete and correct. If you have cited papers that have been retracted, please include the rationale for doing so in the manuscript text, or remove these references and replace them with relevant current references. Any changes to the reference list should be mentioned in the cover letter that accompanies your revised manuscript.

We expect to receive your revised manuscript within two weeks. 

**DATA POLICY**

Thank you for submitting all the data underlying the graphs shown in the figures.

Please be aware that the data that you have deposited in the GenBank GEO database NCBI (GSE168483) should be made publicly available at this time, before the manuscript enters production.

*Published Peer Review History*

*Early Version*

Best wishes,

Ines

--

Ines Alvarez-Garcia, PhD,

Senior Editor,

ialvarez-garcia@plos.org,

PLOS Biology

Reviewers' comments:

Rev. 1:

The authors did a good job in addressing the reviewer concerns that were highlighted by the editor.

In relation to the characterisation of the immune cell populations present in the various segments, the markers that the authors selected to identify the different cell types were not very conventional. For example CTLA4 for Tregs instead of Foxp3 and, the combination of Itgax/C1qc for DCs and macrophages. In this last example, the reference 20 does not support the use of Itgax/C1qc for macrophage/DC identification so a more appropriate reference should be added.

Rev. 2:

Has not submitted any comments but recommends acceptance.

---

## [Editor Report · Decision Letter 3]

1 Oct 2021

Dear Shalev,

On behalf of my colleagues and the Academic Editor, Emma Rawlins, I am pleased to say that we can in principle offer to publish your Short Reports entitled "Spatial gene expression maps of the intestinal lymphoid follicle and associated epithelium identify zonated expression programs" in PLOS Biology, provided you address any remaining formatting and reporting issues. These will be detailed in an email that will follow this letter and that you will usually receive within 2-3 business days, during which time no action is required from you. Please note that we will not be able to formally accept your manuscript and schedule it for publication until you have made the required changes.

PRESS

Sincerely, 

Ines

--

Ines Alvarez-Garcia, PhD 

Senior Editor 

PLOS Biology
